# Shear Wave Elastography to Guide Perineural Hydrodissection: Two Case Reports

**DOI:** 10.3390/diagnostics10060348

**Published:** 2020-05-28

**Authors:** Daniel Chiung-Jui Su, Ke-Vin Chang, Stanley K. H. Lam

**Affiliations:** 1Department of Physical Medicine and Rehabilitation, Chi-Mei Medical Center, Tainan 710, Taiwan; dr.daniel@gmail.com; 2Department of Physical Medicine and Rehabilitation, National Taiwan University Hospital, Bei-Hu Branch, Taipei 108, Taiwan; kvchang011@gmail.com; 3Department of Family Medicine, Faculty of Medicine, The Chinese University of Hong Kong, Sha Tin 999077, Hong Kong

**Keywords:** humeral fracture, nerve tissue regeneration, osteosynthesis, fracture, radial nerve, scarring, ultrasound imaging

## Abstract

Radial nerve palsy is not uncommon after humeral shaft fractures. Ultrasound-guided hydrodissection is an emerging treatment for nerve entrapment. Two cases of radial nerve injury after humeral shaft fractures with plate fixation are presented. Shear wave elastography was used to identify hardened scars surrounding the nerve, causing entrapment. These areas were marked on the skin as targets for ultrasound-guided hydrodissection. Each patient experienced full recovery of their radial nerve function. Shear wave elastography may be used to precisely identify sites of neural entrapment by scar tissue and accurately guide perineural hydrodissection, particularly in complex postoperative cases.

## 1. Introduction

Perineural hydrodissection is an emerging technique in the treatment of chronic neuropathic pain caused by nerve entrapment [1]. It is a minimally invasive technique that uses the high-pressure injection of fluid to dissect the nerve from the surrounding tissue and remove adhesions, which allows the entrapped nerve to glide smoothly again. When performing hydrodissection, it is critical to identify the exact points of entrapment in order to ensure the maximal effectiveness of the procedure and prevent a recurrence. Physicians use high-resolution ultrasound (US) to scan the nerve from proximal to distal and identify sudden increases in its cross-sectional area (CSA), the notch sign, any enlarged fascicles, and hypoechoic changes in the fascicles or hyperechoic scars encasing them [2]. 

After identifying the entrapments, perineural hydrodissection can be performed under ultrasound guidance. However, in complicated cases like postsurgical neuropathy, the scar tissues usually extend from the superficial to the deep fascia and have multiple sites of involvement with the nerve. The anatomical planes may also be disrupted, which further increases the difficulty of identifying the entrapment sites that need the intervention the most.

Shear wave elastography (SWE) is an emerging US imaging technique. By transmitting longitudinal acoustic pulses, it leads to tissue displacement, resulting in the propagation of shear waves away from the region of excitation. By determining the shear wave velocity (SWV), SWE facilitates the identification and quantification of tissue hardness [3]. We hypothesized that this technique may be useful for identifying the precise sites of scar-related neural compression in complex postoperative cases where conventional US guidance is limited due to anatomic distortion. 

Herein, we describe two cases of postoperative radial nerve entrapment following humeral fracture fixation. We highlight the utility of SWE in identifying perineural scarring along the course of the nerve, facilitating targeted hydrodissection, and assessing resolution on follow-up examinations.

## 2. Case Presentations

Informed consent has been obtained according to guidelines from the Helsinki declaration and research ethics. 

Case 1: A 31-year-old man sustained a left humeral shaft fracture in a traffic accident and underwent fixation with a dynamic compression plate. Fourteen months later, the plate was removed, after which the patient developed a wrist drop of his left hand. Neurodiagnostic studies (i.e., nerve conduction velocity and electromyography) were performed two months after the surgery and showed severe axonal damage with no signals from the radial nerve. Spontaneous activity was noted over the supinator, extensor carpi radialis longus (ECRL), extensor digitorum communis (EDC), and extensor indicis proprius (EIP) muscles, without any motor unit action potentials (MUAP) during recruitment. The ultrasound (US) examination showed a continuous radial nerve with an increased CSA and hypoechoic changes to the nerve fascicles at both the entrance to the spiral groove and supinator levels. Our US system (Acuson S3000, Siemens Healthcare (Pty) Ltd., Erlangen, Germany) entails a linear-array transducer with a bandwidth of 4–9 MHz. The SWV of the radial nerve was assessed using dedicated software (Virtual Touch Tissue Imaging Quantification (VTIQ), Siemens Healthcare (Pty) Ltd., Erlangen, Germany). With the VTIQ software, SWV can be assessed from 0–10 m/s. Segment-by-segment SWE was performed over the course of the radial nerve in the left upper arm, and increased hardness (SWV = 9.94 m/s) was identified at the entrance to the spiral groove, where the nerve starts to wrap around the humerus (Figure 1). The SWV of the contralateral side was 4.05 m/s (Figure 2).

In addition to his motor weakness, the patient also presented with severe neuropathic pain along the radial nerve innervated areas with a visual analog scale (VAS) score of 8 out of 10 immediately after the dynamic plate was removed. In view of the intolerable neuropathic pain and persistent motor weakness below the supinator level after two months, we decided to perform US-guided perineurial hydrodissection with 5% dextrose water (D5W, no lidocaine added) over the point where the SWV had shown hardness. The patient lay on the bed in the right lateral position. The affected limb rested on the side of the body with the elbow flexed at approximately 10° s. We then placed the transducer softly on the skin without applying pressure and examined the SWE. Segments where SWV had shown increased hardness were marked and numbered sequentially from proximal to distal. The numbers were matched with the SWE images on the US machine for future comparison. Photos of the marks on the skin were taken from lateral to medial. The photos were used during each hydrodissection so that the SWE measurement did not need repeating, unless some uncertainty arose during subsequent treatments. A 25-gauge needle was used to inject D5W to separate the nerve from the surrounding tissue that was marked as hardened, with the aim of separating the hardened area that was identified in the SWE. Approximately 15 cc of D5W was used each time. The procedure was performed three times at two-week intervals. The SWV, VAS, and muscle strength were documented each time before and immediately after the intervention. 

Immediately after the hydrodissection, the SWV was decreased temporarily due to the artifact of the injectate that infiltrated the soft tissue. The neuropathic pain was decreased from a VAS score of 8 to 2, and the patient experienced a sensation of local fullness. There was no muscle-power change immediately after the hydrodissection. When followed up two months after the last hydrodissection, the neuropathic pain had decreased from a VAS score of 8 to 3 (out of 10). The muscle power of the wrist extensors had considerably improved from 1 to 4 (on a scale of 0 to 5, with 5 being normal), and that of the finger extensors improved from 1 to 3. Neurodiagnostic studies revealed MUAPs over the supinator, ECRL, and EDC muscles, but MUAP remained absent over the EIP. US of the radial nerve still showed swelling and hypoechoic changes in the fascicles, as had been observed prior to treatment. However, the hardness of the surrounding tissue had decreased from 10.0 m/s to 7.45 m/s and in some areas even to 6 m/s (Figure 3, green box). The radial nerve hardness had decreased from 9.94 m/s to 8.62 m/s (Figure 3, yellow box).

Three months later, the neuropathic pain was completely resolved, the muscle power of the ECRL and EDC had increased to 4+ (near normal), and the EIP had increased to 4; the neurodiagnostic study showed reinnervation over the EIP as well. The SWE revealed a further decrease in the hardness to 4.67 m/s at the previous entrapment site at the entrance to the spiral groove (Figure 4, yellow box). No further treatment was required, and the patient had regained full motor function with no numbness at the last follow-up 21 months after the hydrodissection. 

Case 2: A 43-year-old man suffered a left humeral shaft fracture and underwent plate fixation. He sustained a postoperative radial nerve palsy and presented to our institution one year after the previous surgery. US confirmed that the radial nerve was compressed underneath the plate. Therefore, removal of the plate with neurolysis of the radial nerve was performed. One year after removal of the plate, the motor function of the radial nerve had returned to normal, but the patient now presented with severe tenderness and allodynia over the cutaneous innervation areas of the posterior cutaneous nerve of forearm and the superficial radial nerve. 

Neurodiagnostic studies showed moderate axonal damage to the radial nerve, and there was no signal over the posterior cutaneous nerve of forearm or over the superficial radial nerve. There was no spontaneous activity, but reinnervation signals were present over the supinator, ECRL, EDC, and EIP.

US revealed hypoechoic changes over the radial nerve from its entrance to its exit from the spiral groove. The posterior cutaneous nerve of forearm and superficial radial nerve were both swollen and hypoechoic after their respective exits from the groove. The biceps and triceps muscles and the lateral intermuscular septum were surrounded by scar tissue, were hyperechoic, and demonstrated a loss of the normal fibrillary pattern under the ultrasound.

We performed cross-sectional SWE of the radial nerve and its branches along their course from the entrance into the spiral groove to the elbow level. Points of increased hardness, as high as 9.91 m/s, were identified with SWV at the spiral groove level. (Figure 5). Although some of the cross-section views of the radial nerve and the posterior cutaneous nerve of forearm looked swollen and hypoechoic after their respective exits from the spiral groove, there was no increased hardness measured with SWE (Figure 6).

The SWV of the posterior cutaneous nerve of forearm was 3.48 m/s (Figure 6). This indicated that, although the patient had allodynia over the innervation area of the posterior cutaneous nerve of the forearm, the nerve was injured proximal to its exit from the spiral groove. Multiple segments of increased hardness along the radial nerve and its branches were marked with a marker on the skin before we performed ultrasound-guided hydrodissection (Figure 7a). Five to ten cubic centimeters of D5W were used in each of the mark-up segments, which added up to a total of 60 cc of D5W to dissect the adhered scar tissues around the nerves. 

After six hydrodissection sessions at two-week intervals, the hyperesthesia along the superficial radial nerve and posterior cutaneous nerve innervation areas on the forearm were markedly improved, and the previously observed erythema over the affected area had also returned to normal (Figure 7b). The superficial operation scar itself also appeared less prominent due to the hydrodissection effect underneath it. The patient had no numbness or allodynia at the last follow-up, two years after the last hydrodissection. 

## 3. Discussion

A radial nerve injury in humeral fractures can occur either at the time of the injury, during plate fixation, during osteosynthesis, or at the time of plate removal. A primary injury happens at the time of the fracture, when the nerve is usually either crushed or stretched. A secondary injury occurs during surgery if open reduction and fixation is required. A tertiary injury may result when scar tissue starts to grow, causing fascia adhesion and direct nerve tethering or even encasement. Scar tissue may also limit fascial gliding during limb movements and entrap the nerve where it passes from one fascial plane to the other. Cattin et al. [4] demonstrated that, following axonal damage to the nerves, macrophages are the predominant cells that clear the axonal and myelin debris. When scar tissue forms around the nerve ends, this debris-clearing mechanism is jeopardized, and axonal regrowth is hindered. Therefore, the amount of scar tissue formation plays an important role in determining whether nerve regeneration will be successful. 

Traditionally, when using US to identify entrapment sites, we either look for sudden changes in the CSA of the nerve, the notch sign, any enlarged fascicles, hypoechoic changes in the fascicles, and hyperechoic signals surrounding the nerve, or we try to reproduce symptoms with sonopalpation [2,5]. However, we must be aware that scar tissue can grow in several areas along the course of a nerve, causing multiple entrapments. The distorted anatomy of the tissue/neural architecture after the operation makes identification of precise scar tissues surrounding the nerve difficult with conventional US. As in our second patient, scar tissue may encroach upon the nerve from the bone, causing adhesion (Figure 6), and the hyperechoic bony cortex makes identification of the precise location of the scar with conventional US techniques more difficult. 

In the second patient, SWE demonstrated that the size and echogenicity of a nerve segment do not necessarily reflect the hardness of the surrounding tissues. Whole segments of the nerve may be swollen in severe cases, which makes identification of the precise location of injury with conventional US techniques difficult. However, identifying these hardened areas is critical for successful perineural hydrodissection. Using SWE to guide perineural hydrodissection can avoid unnecessary treatment of nerve segments which appear abnormal on US due to prior surgery, but which are not surrounded by scar tissue and do not contribute to symptoms.

In our patients, we used SWE with VTIQ software. Unlike strain elastography, the SWV measured with SWE is an absolute value that can be compared between individuals [6,7]. The harder the tissue, the faster the SWV. Compared to strain elastography using external force or using internal mechanical stimulus, SWE is considered more objective and less influenced by interoperator variability [8]. In a systematic review and meta-analysis study, Lin et al. found that SWE may be more sensitive than strain imaging in discriminating between patients with carpal tunnel syndrome and those without [9]. Therefore, scar lesions that can be detected by SWE may not be detected by strain elastography.

In addition, previous studies regarding the use of US elastography for the evaluation of peripheral nerves mainly focused on the median nerve at the carpal tunnel level [8]. In the carpal tunnel, many tendons are situated close to the median nerve, which may contaminate the results and increase the difficulty of measuring the hardness. The difficulty of measurement is further increased in patients who have undergone surgery. We examined the radial nerve at the upper arm in our patients, where there is no tendon close by, except for pennate muscles like triceps and brachialis. Therefore, the artifact of the tendon did not influence our cases. Additionally, according to Kentaro et al., the influence of the pennation angle on the measurement of the shear wave velocity of pennate muscles is negligible [10]. This means that the effect of transducer orientation in this region when assessing the SWE of the radial nerve is minimal. By combining the SWE data from longitudinal and axial orientations, performers can create a three-dimensional hardness map of the area of interest. 

There are some limitations to SWE. SWE is still an emerging technique and is operator-dependent, like any US scanning. Many brands of US machine have an SWE function, but each creates the shear wave and the data being analyzed differently [11]. There are reported artifacts in supersonic shear wave elastography (Version 6.2, Aixplorer, SuperSonic Imagine, Aix-en-Provence, France) [12] due to the design of the transducer. A bone-proximity artifact was reported in a study evaluating the median nerve with SWE using a Toshiba Aplio 500, Toshiba, Tokyo, Japan [13]. Currently, there are no reports of known technical artifacts linked to the US machine we use, and, in our patients, we compared both upper limbs to make sure there were no, or minimal, bone-proximity artifacts. Lastly, SWE studies often display results in Young’s modulus (E), by converting SWV to E using the equation E = 3pv^2^ where v is SWV and p is tissue density. By contrast, in our patients with postoperative scar tissues, the tissue density was not a constant value. Therefore, values in this study were reported as SWV (m/s).

## 4. Conclusions

SWE may be used to precisely identify sites of nerve entrapment by perineural scar tissue and to accurately guide targeted hydrodissection. The technique may be of particular benefit in complex postoperative cases, and further prospective controlled studies are required to elucidate its role and utility in this clinical scenario.

## Figures and Tables

**Figure 1 diagnostics-10-00348-f001:**
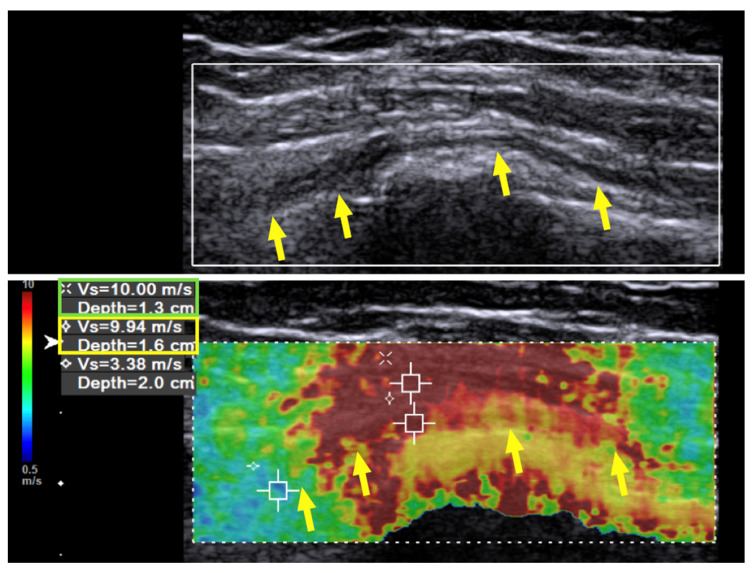
Case 1: Shear wave velocity before hydrodissection. Shear wave velocity (SWV) investigation of the radial nerve over the spiral groove before hydrodissection in a 31-year-old man with wrist drop following plate removal after humerus fracture osteosynthesis. Multiple regions of interest are selected, including the radial nerve and the surrounding tissues. The SWV of the radial nerve is 9.94 m/s and that of the surrounding soft tissue is 10 m/s. Yellow arrow and box: radial nerve in longitudinal view. Green box: scar tissue.

**Figure 2 diagnostics-10-00348-f002:**
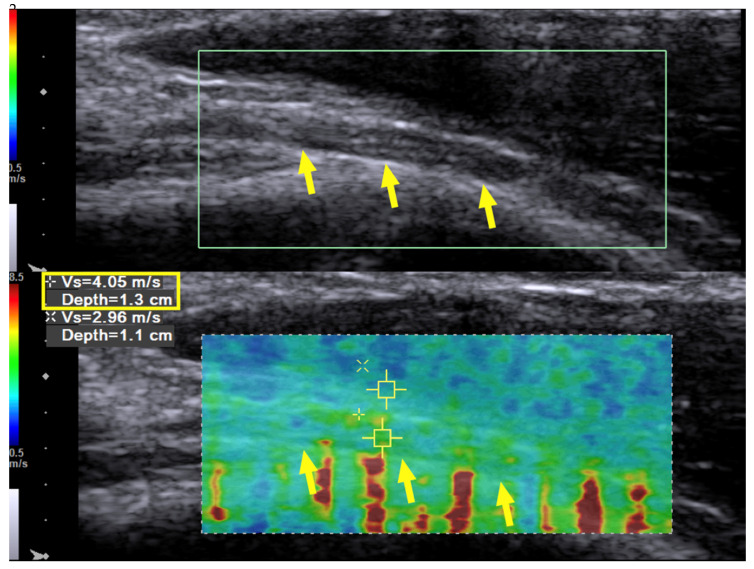
Case 1: Ultrasound imaging with shear wave elastography. Long-axis ultrasound imaging of the radial nerve on the contralateral arm with shear wave elastography. Shear wave velocity measurement of the normal radial nerve was 4.05 m/s (longitudinal view at the beginning of the spiral groove). Yellow arrow and box: radial nerve in longitudinal view.

**Figure 3 diagnostics-10-00348-f003:**
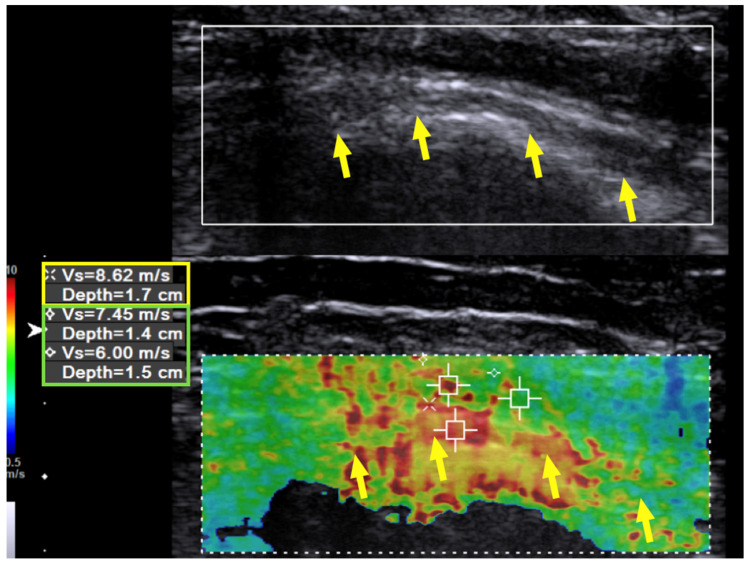
Case 1: Shear wave velocity after hydrodissection. Shear wave velocity (SWV) investigation of the radial nerve (long axis) over the spiral groove after triple hydrodissection in a 31-year-old man with wrist drop following plate removal after humerus fracture osteosynthesis. After triple hydrodissection, the hardness of the scar (green box) and radial nerve (yellow box) over the spiral groove has decreased. Yellow arrow: radial nerve in longitudinal view.

**Figure 4 diagnostics-10-00348-f004:**
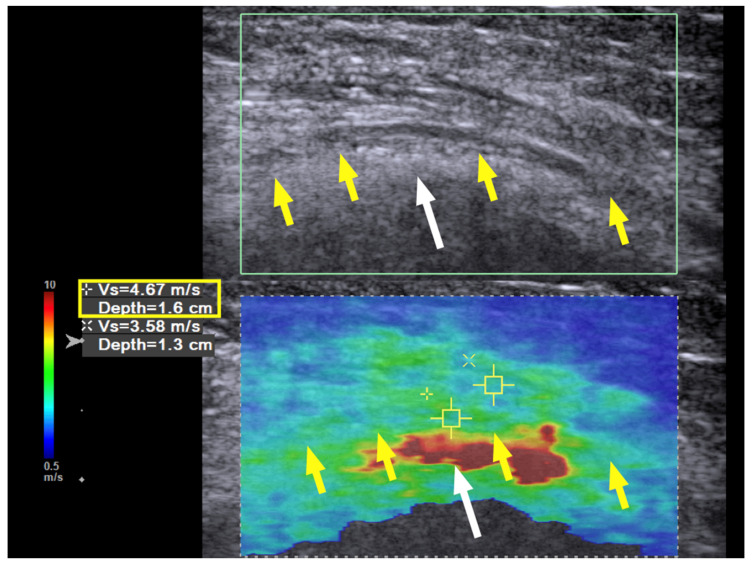
Case 1: Shear wave velocity nine months after hydrodissection. Shear wave velocity (SWV) investigation of the radial nerve (long axis) over the spiral groove. Nine months after the first hydrodissection, the hardness of the nerve has returned to near normal, with a shear wave velocity of 4.67 m/s. Yellow arrow and box: radial nerve in longitudinal view; white arrow: bony cortex of humerus.

**Figure 5 diagnostics-10-00348-f005:**
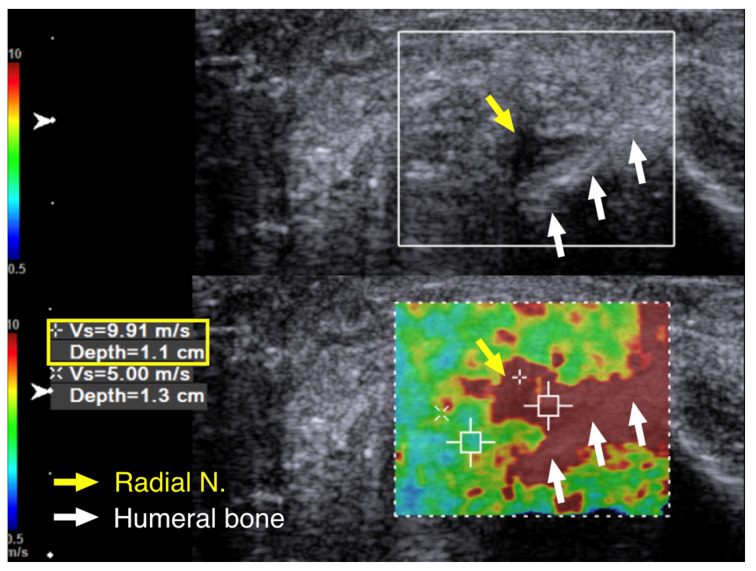
Case 2: Shear wave velocity at the spiral groove level. A 43-year-old man with radial nerve palsy after plating of a left humeral shaft fracture. Cross-sectional ultrasound imaging of the radial nerve with shear wave elastography, with a shear wave velocity of 9.91 m/s. Yellow arrow and box: radial nerve in cross -sectional view. White arrow: humeral bone.

**Figure 6 diagnostics-10-00348-f006:**
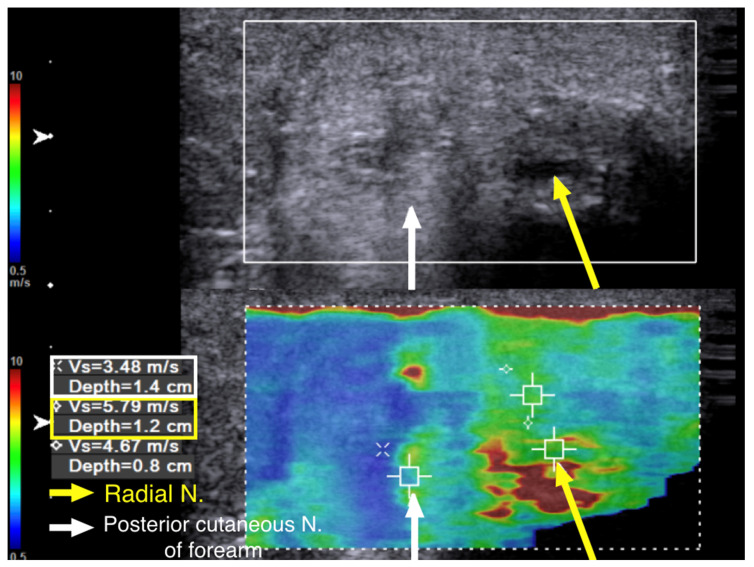
Case 2: Shear wave velocity at the exit of the spiral groove. A 43-year-old man with radial nerve palsy after plating of a left humeral shaft fracture. Ultrasound images with cross-sectional view of the radial nerve after its exit from the spiral groove. The radial nerve and the posterior cutaneous nerve of the forearm were swollen. The shear wave velocity of the radial nerve was 5.79 m/s (yellow arrow and box), and that of the posterior cutaneous nerve of the forearm was 3.48 m/s (white arrow and box).

**Figure 7 diagnostics-10-00348-f007:**
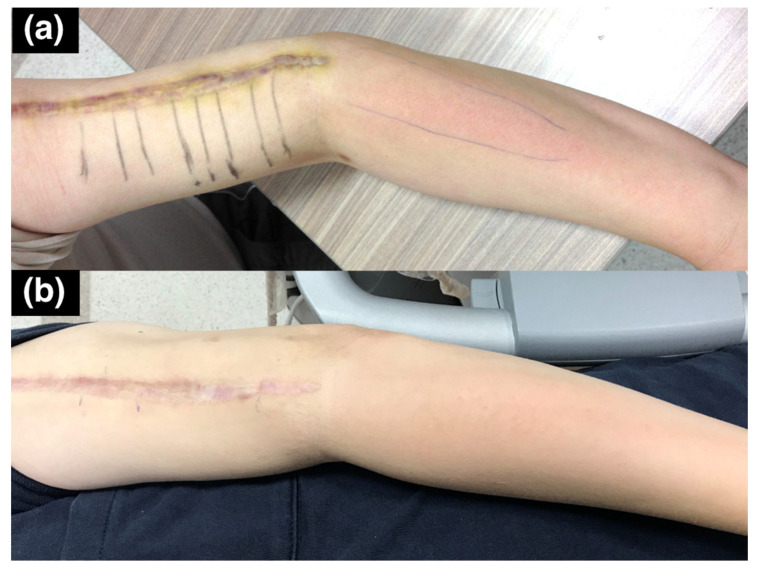
Case 2: Targets for hydrodissection. A 43-year-old man with radial nerve palsy after plating of a left humeral shaft fracture. (**a**) Before hydrodissection. Marks indicate the cross-sections with increased shear wave velocity during the shear wave elastography for injection. There is erythema over the innervation area of the posterior cutaneous nerve of the forearm. (**b**) After six hydrodissection sessions, the erythema improved and the allodynia disappeared.

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
