# Peer review of "Shear Wave Elastography to Guide Perineural Hydrodissection: Two Case Reports"

_diagnostics, 2020, doi:10.3390/diagnostics10060348_

Round 1

Reviewer 1 Report

This paper is well written and acceptable in present form.

Author Response

I have revised the manuscript according to the reviewer's advice. Please see the attachment. Thanks!

Reviewer 2 Report

Summary: the authors report 2 cases of post-operative radial nerve entrapment neuropathy after plate osteosynthesis of humeral shaft fractures (one after plate removal; one which occurred during original ORIF). They describe how the sites of nerve entrapment were identified using SWE and treated using perineural hydrodissection with D5W, and provide follow-up details. They discuss causes of radial neuropathy, mechanisms of scarring and nerve entrapment and regrowth, and provide some information as to why they use the technique and in which situations it may be useful. They conclude that the technique is useful to identify sites of neural entrapment to guide hydrodissection.

Major Strengths:

  1. Timely and original subject matter. I believe this may be the first report documenting the use of SWE guidance for perineural hydrodissection, and it is certainly the first for this specific clinical scenario. SWE is a rapidly evolving modality and this manuscript adds useful information to the literature regarding its potential utility for post-operative radial nerve palsy.
  2. Generally well written and readable manuscript

***Major Weaknesses:

  1. The Case presentations section does not include sufficient details of the US machine and techniques used (the authors have misplaced this information in the discussion section). Details of the perineural dissection technique are lacking and readers could not reproduce the technique based on the information provided by the authors. The authors do not describe immediate outcomes (especially pain relief) following hydrodissection, and follow-up details in Case 2 are insufficient.
  2. The Discussion section is weak. The authors spend a long time discussing causes and mechanisms of radial entrapment neuropathy, but fail to explain clearly their reason for using SWE (i.e. its advantages) over strain elastography or conventional US. Almost an entire paragraph should be in the case presentations (method) section. The authors have not discussed drawbacks of the technique nor limitations of their study.

Minor Weaknesses

  1. The introduction needs minor/moderate re-organisation and text editing to improve organisation and readability.
  2. Few minor comments and optional additional information or rephrasing to enhance impact of language

Specific comments by section:

Title: OK

Abstract: The last sentence could be more precise. e.g. "SWE may be used to precisely identify sites of neural entrapment by scar tissue, and accurately guide perineural hydrodissection, particularly in complex post-operative cases"

Introduction:

Paragraph 1 Line 1 (P1L1) - "widely used"... perhaps in the authors' institution, but not worldwide. I would rephrase as "emerging technique" for an international audience

*Paragraphs 3 could be improved by reorganising the text. Currently the authors disrupt the flow of ideas by positioning technical details in the last 2 sentences. The authors could keep the first 2 sentences --> then mention that By determining SWV, SWE facilitates identification and quantification of tissue hardness (technical details could be included in parentheses) --> then mention that they hypothesised that the technique may be useful to identify the precise sites of scar-related neural compression in complex post-operative cases where conventional US guidance is limited due to anatomic distortion.

*Paragraph 4 could also be improved. Consider moving consent statement to Case presentations section (before the 2 cases). Mention which guidelines were used - institutional review board? Helsinki declaration? The last sentence needs to be more precise: it should state what the authors have done ("herein we describe 2 cases of postoperative radial nerve entrapment following humeral fracture fixation), and what they are showing ("we highlight the utility of SWE in identifying perineural scarring along the course of the nerve, facilitating targeted hydrodissection, and assessing resolution on follow-up examinations") - or similar comments

Case Presentations

Include consent details here.

Case 1:

*P1L8 and P1L10 - I would suggest including details of US machine, probe frequency, cursor placement and software here in the "methods" section and not in the discussion as is currently the case

*P2L1 - first sentence fits better at end of P1. When did the patient develop neuropathic pain - immediately post-op or delayed (i.e. after onset of muscle weakness)

***P2L6 - The authors MUST specify more details about how the procedure was performed here. Details about skin photography/marking should be here. What volume of D5W was used? Was lidocaine used? What needle gauge? What was the endpoint - fixed volume, US appearances, or symptoms?

*P3L1-3 - was there any immediate effect on analgesia, muscle power or SWV after hydrodissection? D5W is known to have rapid acting anaesthetic properties. Any immediate complications? What system of muscle grading was used (MRC?) - please specify.

P3L4 - Keep the same structure throughout the text: clinical findings, muscle power, NCS, then US. You have mentioned NCS after US in the current version of your paper.

P4 - I assume pain had completely resolved at this point of follow-up? Consider specifying this so it is clear to the readership.

Case 2

P1L2 - specify "post-operative" radial nerve palsy

P1L3 - avoid vague statements. How was it established that the Radial nerve was compressed by the DCP? Clinical? US? Be precise.

P2 - again it's helpful for readers if you are precise. Is this a conduction block or what degree of axonal injury is it? The first case was obviously severe, but this case?

P3L4 - need to insert "demonstrated" before ...loss of....

*P6 - what was the time interval between sessions (2weeks again? Please specify). How long was the follow-up after the final session? What is the patient's current status at last available follow-up? The scar appears considerably less prominent in 7b than 7a - is this photographic "artifact" or a real effect of hydrodissection, or healing?

***Discussion

Unfortunately this section lets the manuscript down in my opinion. I would suggest that you restructure the text as follows:-

  • Brief discussion of radial nerve injury mechanisms and pathology, including the effects of scar tissue on nerve regeneration and function (gliding). This information is largely present in P1 and P3, but these are far too long and should be combined into a shorter, more precise paragraph
  • Discussion of how important it is to precisely identify sites of compression since these may be multiple (you could include information from P5 here) and difficult to evaluate on conventional US due to anatomic distortion of tissue/neural architecture.
  • Comment on your institutional practice (as you have done in P4) but indicate WHY you use SWE - i.e. what are the advantages and how are these illustrated by the cases. To my mind, these are: (i) ability to correlate conventional US appearances of neural entrapment with tissue hardness measurement to precisely identify sites causing symptoms; identify multiple potential entrapment sites in complex post-operative patients; and avoid unnecessary treatment of nerve segments which appear abnormal on US due to prior surgery, but whichare not surrounding by scar tissue and are not contributing to symptoms (Case 2); (ii) more sensitive assessment of resolution of scarring even when conventional US appearances are static (Case 1) using quantitative assessment (unlike strain elastography); (iii) Excellent efficacy of treatment in these 2 cases.
  • Brief comments on limitations of technique (operator dependent, bone artifacts, limited penetration); limitations of case interpretation (cannot disentangle hydrodissection vs. D5W direct anti-neurogenic inflammatory effects; variable follow-up; any confounding therapies (e.g. physiotherapy or other scar treatment?), possibility that patients may have recovered spontaneously without treatment).

Conclusion - OK but could be written more succinctly

e.g. SW may be used to precisely identify sites of nerve entrapment by perineural scar tissue and accurately guide targeted hydrodissection. The technique may be of particular benefit in complex post-operative cases, and further prospective controlled studies are required to elucidate its role and utility in this clinical scenario.

Figures - OK

References - OK, authors may wish to include doi: 10.1213/XAA.0000000000001143 as this is the only similar paper in the literature

Author Response

Thanks for your comment. I have revised the manuscript according to your advice.

Specific comments by section:

Title: OK

Abstract: The last sentence could be more precise. e.g. "SWE may be used to precisely identify sites of neural entrapment by scar tissue, and accurately guide perineural hydrodissection, particularly in complex post-operative cases”

I have changed the last sentence as indicated. 

Introduction:

Paragraph 1 Line 1 (P1L1) - "widely used"... perhaps in the authors' institution, but not worldwide. I would rephrase as "emerging technique" for an international audience

I have rephrased it as “an emerging technique”.

*Paragraphs 3 could be improved by reorganising the text. Currently the authors disrupt the flow of ideas by positioning technical details in the last 2 sentences. The authors could keep the first 2 sentences --> then mention that By determining SWV, SWE facilitates identification and quantification of tissue hardness (technical details could be included in parentheses) --> then mention that they hypothesised that the technique may be useful to identify the precise sites of scar-related neural compression in complex post-operative cases where conventional US guidance is limited due to anatomic distortion.

*Paragraph 4 could also be improved. Consider moving consent statement to Case presentations section (before the 2 cases). Mention which guidelines were used - institutional review board? Helsinki declaration? The last sentence needs to be more precise: it should state what the authors have done ("herein we describe 2 cases of postoperative radial nerve entrapment following humeral fracture fixation), and what they are showing ("we highlight the utility of SWE in identifying perineural scarring along the course of the nerve, facilitating targeted hydrodissection, and assessing resolution on follow-up examinations") - or similar comments

I have rewrote the introduction, please see the attachment. 

Case Presentations

Include consent details here.

I have already added consent details. 

Case 1:

*P1L8 and P1L10 - I would suggest including details of US machine, probe frequency, cursor placement and software here in the "methods" section and not in the discussion as is currently the case

*P2L1 - first sentence fits better at end of P1. When did the patient develop neuropathic pain - immediately post-op or delayed (i.e. after onset of muscle weakness)

***P2L6 - The authors MUST specify more details about how the procedure was performed here. Details about skin photography/marking should be here. What volume of D5W was used? Was lidocaine used? What needle gauge? What was the endpoint - fixed volume, US appearances, or symptoms?

*P3L1-3 - was there any immediate effect on analgesia, muscle power or SWV after hydrodissection? D5W is known to have rapid acting anaesthetic properties. Any immediate complications? What system of muscle grading was used (MRC?) - please specify.

P3L4 - Keep the same structure throughout the text: clinical findings, muscle power, NCS, then US. You have mentioned NCS after US in the current version of your paper.

P4 - I assume pain had completely resolved at this point of follow-up? Consider specifying this so it is clear to the readership.

Case 2

P1L2 - specify "post-operative" radial nerve palsy

P1L3 - avoid vague statements. How was it established that the Radial nerve was compressed by the DCP? Clinical? US? Be precise.

P2 - again it's helpful for readers if you are precise. Is this a conduction block or what degree of axonal injury is it? The first case was obviously severe, but this case?

P3L4 - need to insert "demonstrated" before ...loss of....

*P6 - what was the time interval between sessions (2weeks again? Please specify). How long was the follow-up after the final session? What is the patient's current status at last available follow-up? The scar appears considerably less prominent in 7b than 7a - is this photographic "artifact" or a real effect of hydrodissection, or healing?

***Discussion

Unfortunately this section lets the manuscript down in my opinion. I would suggest that you restructure the text as follows:-

  • Brief discussion of radial nerve injury mechanisms and pathology, including the effects of scar tissue on nerve regeneration and function (gliding). This information is largely present in P1 and P3, but these are far too long and should be combined into a shorter, more precise paragraph
  • Discussion of how important it is to precisely identify sites of compression since these may be multiple (you could include information from P5 here) and difficult to evaluate on conventional US due to anatomic distortion of tissue/neural architecture.
  • Comment on your institutional practice (as you have done in P4) but indicate WHY you use SWE - i.e. what are the advantages and how are these illustrated by the cases. To my mind, these are: (i) ability to correlate conventional US appearances of neural entrapment with tissue hardness measurement to precisely identify sites causing symptoms; identify multiple potential entrapment sites in complex post-operative patients; and avoid unnecessary treatment of nerve segments which appear abnormal on US due to prior surgery, but whichare not surrounding by scar tissue and are not contributing to symptoms (Case 2); (ii) more sensitive assessment of resolution of scarring even when conventional US appearances are static (Case 1) using quantitative assessment (unlike strain elastography); (iii) Excellent efficacy of treatment in these 2 cases.
  • Brief comments on limitations of technique (operator dependent, bone artifacts, limited penetration); limitations of case interpretation (cannot disentangle hydrodissection vs. D5W direct anti-neurogenic inflammatory effects; variable follow-up; any confounding therapies (e.g. physiotherapy or other scar treatment?), possibility that patients may have recovered spontaneously without treatment).

I have rewritten the discussion per your suggesions. Please see the attachment. 

Conclusion - OK but could be written more succinctly

e.g. SW may be used to precisely identify sites of nerve entrapment by perineural scar tissue and accurately guide targeted hydrodissection. The technique may be of particular benefit in complex post-operative cases, and further prospective controlled studies are required to elucidate its role and utility in this clinical scenario.

Figures - OK

References - OK, authors may wish to include doi: 10.1213/XAA.0000000000001143 as this is the only similar paper in the literature

I have added references of several papers regarding hydrodissection and SWE. 

Reviewer 3 Report

interesting case report on a clever use of SWE. It is a report of an interesting observation, using SWE to locate adhesions for subsequent neurolysis. Writing is fine.

Author Response

Thanks for your comment. I have revised the manuscript according to the reviewer's advice. 

Round 2

Reviewer 2 Report

The authors have taken on board most of my suggestions and have vastly improved the introduction and case presentation sections in particular. These sections now read quite well.

Minor suggestions for Discussion section:

  1. Case 2 L3 "US confirmed that the radial nerve was compressed underneath the plate by US scanning. " This sentence needs to be checked for grammar.
  2. P4 L1 - check grammar - should read "we used SWE with VTIQ software"
  3. P4 L3 - it would be much better to delete  "SWE provides a more objective and consistent tissue compres- sion than does strain elastography. [6]."  and replace it (i.e. cut and paste) with P6 L2-7 "Compared to strain elastography using external force or using internal mechanical stimulus, SWE is considered more objective and less influenced by inter-operator varia- bility. [6, 7, 8]. In a systematic review and meta- analysis study, Lin et al. found that SWE may be more sensitive than strain imaging in discriminating between patients with CTS and those with- out[10]. Therefore, scar lesions that can be detected by SWE may not be detected by strain elas- tography."

3. After making the above change in (2), the authors should combine P6-8 into one section discussing limitations.

Once the authors have done this, I think the manuscript will be well written, much better organised, more informative for readers, and entirely suitable for publication. I congratulate the authors for their hard work in revising the methods section in particular, and in submitting a manuscript with the potential to significantly change practice.

Author Response

Minor suggestions for Discussion section:

  1. Case 2 L3 "US confirmed that the radial nerve was compressed underneath the plate by US scanning. " This sentence needs to be checked for grammar.

I have corrected the grammar of the sentence. 

    2.  P4 L1 - check grammar - should read "we used SWE with VTIQ  software"

I have checked the grammar accordingly. 

    3.  P4 L3 - it would be much better to delete  "SWE provides a more objective and consistent tissue compres- sion than does strain elastography. [6]."  and replace it (i.e. cut and paste) with P6 L2-7 "Compared to strain elastography using external force or using internal mechanical stimulus, SWE is considered more objective and less influenced by inter-operator varia- bility. [6, 7, 8]. In a systematic review and meta- analysis study, Lin et al. found that SWE may be more sensitive than strain imaging in discriminating between patients with CTS and those with- out[10]. Therefore, scar lesions that can be detected by SWE may not be detected by strain elas- tography."

I have revised it as suggested. 

     4.  After making the above change in (2), the authors should combine P6-8 into one section discussing limitations.

I have combined P6-8 into one section.